# On Some Numerical Methods for Solving Large Differential Nonsymmetric Stein Matrix Equations

Lakhlifa Sadek [ID], El Mostafa Sadek *[ID] and Hamad Talibi Alaoui

Laboratory of Engineering Sciences for Energy, National School of Applied Sciences of El Jadida,
University Chouaib Doukkali, Av. des Facultés, El Jadida 24000, Morocco
* Correspondence: sadek.maths@gmail.com

**Abstract:** In this paper, we propose a new numerical method based on the extended block Arnoldi algorithm for solving large-scale differential nonsymmetric Stein matrix equations with low-rank right-hand sides. This algorithm is based on projecting the initial problem on the extended block Krylov subspace to obtain a low-dimensional differential Stein matrix equation. The obtained reduced-order problem is solved by the backward differentiation formula (BDF) method or the Rosenbrock (Ros) method, the obtained solution is used to build the low-rank approximate solution of the original problem. We give some theoretical results and report some numerical experiments.

**Keywords:** extended block Krylov; low-rank approximate solutions; differential Stein matrix equations; BDF method; Rosenbrock method

## 1. Preliminaries

In the present paper, we are interested in the numerical solution of large-scale differential nonsymmetric Stein matrix equations on the time interval $[t_0, T_f]$ of the form:

$$\begin{cases} \dot{X}(t) = AX(t)B - X(t) + EF^T, & t \in [t_0, T_f], \quad (DNSE); \\ X(t_0) = X_0, \end{cases} \quad (1)$$

where $A$ and $B$ are real, sparse and square matrices of size $n \times n$ and $p \times p$, respectively, and $E$ and $F$ are matrices of size $n \times r$ and $p \times r$, respectively, with $r \ll n, p$. The initial condition is given in a factored form as $X_0 = Z_0 \tilde{Z}_0^T \in \mathbb{R}^{n \times p}$ and the matrices $A$ and $B$ are assumed to be large and sparse.

Differential Stein matrix equations play an important role in many problems in control and filtering theory for discrete-time large-scale dynamical systems and other problems; see [1–8]. For solving the large matrix equations during the last years, several projection methods based on Krylov subspaces have been proposed, for example Sylvester matrix equations, differential Sylvester matrix equations, Riccati and others, see, e.g., [8–16]. The main idea employed in these methods is to project the initial problem by using extended Krylov subspaces and then apply the Petrov–Galerkin orthogonality condition.

In this work, we present an extended block Krylov method for solving large differential Stein matrix equations. Our method uses the Petrov–Galerkin projection method and the extended block Arnoldi algorithm. The problem (1) is projected onto small extended Krylov subspaces to obtain low-order differential Stein equations that are solved by a BDF method or Ros method. The approximate solution is then given as a product of matrices with low rank.

We recall the extended block Arnoldi algorithm applied to $(A, V)$ where $V \in \mathbb{R}^{n \times r}$. The extended block Krylov subspace associated to $(A, V)$ (see [8,9,14,15,17]) is given by:

$$\begin{aligned} \mathcal{K}_m^e(A, V) &= Range\{V, A^{-1}V, AV, A^{-2}V, A^2V, \cdots, A^{m-1}V, A^{-m}V\}) \\ &= \mathbb{K}_m(A, V) + \mathbb{K}_m(A^{-1}, A^{-1}V), \end{aligned}$$

where

$$\mathbb{K}_m(A, V) = Range\{V, AV, A^2V, \cdots, A^{m-1}V\}.$$

Algorithm 1 allows us to construct an orthonormal matrix $\mathcal{V}_m = [V_1, V_2, \ldots, V_m]$ that is a basis of the block extended Krylov subspace $\mathcal{K}_m^e(A, V)$. Let $\mathcal{T}_{m,A}$ the restriction of the matrix $A$ to the block extended Krylov subspace $\mathcal{K}_m^e(A, V)$ is given by $\mathcal{T}_{m,A} = \mathcal{V}_m^T A \mathcal{V}_m$. Then, we have the following relations (see [15])

$$A\mathcal{V}_m = \mathcal{V}_m \mathcal{T}_{m,A} + V_{m+1} T_{m+1,m}^A E_m^T.$$

where $E_m = [0_{2r \times 2(m-1)r}, I_{2r}]^T$ and $\mathcal{T}_{m,A} = [T_{i,j}^A]$.

---

**Algorithm 1:** The extended block Arnoldi algorithm (EBA)

---

1. Inputs: $A$ an $n \times n$ matrix, $V$ an $n \times r$ matrix and $m$ an integer.
2. Compute the QR decomposition of $[V, A^{-1}V]$, i.e., $[V, A^{-1}V] = V_1 \Lambda$;
3. Set $\mathcal{V}_0 = [\ ]$;
4. For $j = 1, 2, \ldots, m$

   (a)  Set $V_j^{(1)}$: first $r$ columns of $V_j$; $V_j^{(2)}$: second $r$ columns of $V_j$

   (b)  $\mathcal{V}_j = [\mathcal{V}_{j-1}, V_j]$; $\hat{V}_{j+1} = \left[ A\, V_j^{(1)}, A^{-1} V_j^{(2)} \right]$;

   (c)  Orthogonalize $\hat{V}_{j+1}$ i.e.,

   &ast;   for $i = 1, 2, \ldots, j$

   &ast;   $H_{i,j} = V_i^T \hat{V}_{j+1}$;

   &ast;   $\hat{V}_{j+1} = \hat{V}_{j+1} - V_i H_{i,j}$;

   &ast;   End for

   (d)  Compute the $QR$ decomposition of $\hat{V}_{j+1}$, i.e., $\hat{V}_{j+1} = V_{j+1} H_{j+1,j}$;
5. End For.

---

Throughout the paper, we use the following notations. The Frobenius inner product of the matrices $X$ and $Y$ is defined by $\langle X, Y \rangle_F = tr(X^T Y)$, where $tr(Z)$ denotes the trace of a square matrix $Z$. The associated norm is the Frobenius norm denoted by $\| \|_F$. The Kronecker product $A \otimes B = [a_{i,j}B] \in \mathbb{R}^{np \times np}$ where $A = [a_{i,j}]$. This product satisfies the properties: $(A \otimes B)(C \otimes D) = (AC \otimes BD)$ and $(A \otimes B)^T = A^T \otimes B^T$.

The rest of the paper is organized as follows. In Section 2, we derive the low-rank approximate solutions method and give some theoretical results. In Section 3, we apply the backward differentiation formula method and Rosenbrock method for solving reduced order problem. Section 4 is devoted to some numerical examples.

## 2. Low-Rank Approximate Solutions Method

In this section, we project the large differential equation by using the extended block Krylov subspaces $\mathcal{K}_m^e(A, E)$ and $\mathcal{K}_m^e(B^T, F)$ to obtain low-rank approximate solutions of Equation (1).

We apply the extended block Arnoldi algorithm (Algorithm 1) to the pairs $(A, E)$ and $(B^T, F)$, respectively, and to obtain orthonormal bases $\{V_1, \ldots, V_m\}$ and $\{W_1, \ldots, W_m\}$ and we have

$$\mathcal{T}_{m,A} = \mathcal{V}_m^T A \mathcal{V}_m \text{ and } \mathcal{T}_{m,B} = \mathcal{W}_m^T B^T \mathcal{W}_m,$$

where

$$\mathcal{V}_m = [V_1, V_2, \ldots, V_m] \text{ and } \mathcal{W}_m = [W_1, W_2, \ldots, W_m].$$

We then consider approximate solutions of the large differential Stein matrix Equation (1) that have the low-rank form

$$X_m(t) = \mathcal{V}_m Y_m(t) \mathcal{W}_m^T. \tag{2}$$

The matrix $Y_m(t)$ is determined from the following Petrov–Galerkin orthogonality condition:

$$\mathcal{V}_m^T R_m(t) \mathcal{W}_m = 0, \quad t \in [t_0, T_f], \tag{3}$$

where $R_m(t)$ is the residual

$$R_m(t) = \dot{X}_m(t) - A X_m(t) B + X_m(t) - EF^T. \tag{4}$$

Then, from (4) and (3), we obtain the low dimensional differential Stein equation

$$\begin{cases} \dot{Y}_m(t) = \mathcal{T}_{m,A} Y_m(t) \mathcal{T}_{m,B}^T - Y_m(t) + \widetilde{E}_m \widetilde{F}_m^T, \\ Y_m(t_0) = \mathcal{V}_m X_0 \mathcal{W}_m^T, \end{cases} \tag{5}$$

where $\widetilde{E}_m = \mathcal{V}_m^T E$ and $\widetilde{F}_m = \mathcal{W}_m^T F$.

In the following result, we derive an expression for computation of the norm of the residual R, without having to compute matrix products with the large matrices A and B. This result allows us to reduce the cost in the proposed method when checking if the residual norm is less than some fixed tolerance.

**Theorem 1.** *Let $X_m(t) = \mathcal{V}_m Y_m(t) \mathcal{W}_m^T$ be the approximation obtained at step m by the extended block Arnoldi method, and $Y_m(t)$ the exact solution of low dimensional differential nonsymmetric Stein Equation (5). Then, the residual $R_m(t)$ associated to $X_m(t)$ satisfies the relation*

$$\|R_m(t)\|_F^2 = \|T_{m+1,m}^A \overline{Y}_{m,1}(t) \mathcal{T}_{m,B}^T\|_F^2 + \|T_{m+1,m}^B \overline{Y}_{m,2}(t) \mathcal{T}_{m,A}^T\|_F^2 + \|T_{m+1,m}^A Y_{m,m}(t) (T_{m+1,m}^B)^T\|_F, \tag{6}$$

*where $\overline{Y}_{m,1}(t)$ is the $2r \times 2mr$ matrix corresponding to the last $2r$ rows of $Y_m(t)$, $\overline{Y}_{m,2}(t) = E_m^T Y_m(t)^T$ and $Y_{m,m}(t) = E_m^T Y_m(t) E_m$.*

**Proof.** Using the relations (4) and (2), we have

$$R_m(t) = \mathcal{V}_m \dot{Y}_m(t) \mathcal{W}_m^T - A \mathcal{V}_m Y_m(t) \mathcal{W}_m^T B + \mathcal{V}_m Y_m(t) \mathcal{W}_m^T - EF^T.$$

By using the following relations

$$\begin{cases} A\mathcal{V}_m = \mathcal{V}_{m+1} \begin{bmatrix} \mathcal{T}_{m,A} \\ T_{m+1,m}^A E_m^T \end{bmatrix}, \\ \mathcal{W}_m^T B = \begin{bmatrix} \mathcal{T}_{m,B}^T & E_m(T_{m+1,m}^B)^T \end{bmatrix} \mathcal{W}_{m+1}^T, \\ \mathcal{V}_m = \mathcal{V}_{m+1} \begin{bmatrix} I_{2rm} \\ 0_{2r \times 2r} \end{bmatrix}, \\ \mathcal{W}_m^T = \begin{bmatrix} I_{2rm} & 0_{2r \times 2r} \end{bmatrix} \mathcal{W}_{m+1}^T, \end{cases}$$

we have $R_m(t) = \mathcal{V}_{m+1} \begin{bmatrix} I_{2rm} \\ 0_{2r \times 2r} \end{bmatrix} \dot{Y}_m(t) \begin{bmatrix} I_{2rm} & 0_{2r \times 2r} \end{bmatrix} \mathcal{W}_{m+1}^T - \mathcal{V}_{m+1} \begin{bmatrix} \mathcal{T}_{m,A} \\ T_{m+1,m}^A E_m^T \end{bmatrix}$

$Y_m(t) \begin{bmatrix} \mathcal{T}_{m,B}^T & E_m(T_{m+1,m}^B)^T \end{bmatrix} \mathcal{W}_{m+1}^T + \mathcal{V}_{m+1} \begin{bmatrix} I_{2rm} \\ 0_{2r \times 2r} \end{bmatrix} Y_m(t) \begin{bmatrix} I_{2rm} & 0_{2r \times 2r} \end{bmatrix} \mathcal{W}_{m+1}^T - EF^T,$

since $EF^T = \mathcal{V}_{m+1} \begin{bmatrix} \widetilde{E}_m \widetilde{F}_m^T & 0_{2mr \times 2r} \\ 0_{2r \times 2mr} & 0_{2r \times 2mr} \end{bmatrix} \mathcal{W}_{m+1}^T$, so

$$R_m(t) = \mathcal{V}_{m+1} \begin{bmatrix} \dot{Y}_m(t) - \mathcal{T}_{m,A} Y_m(t) \mathcal{T}_{m,B}^T + Y_m(t) - \widetilde{E}_m \widetilde{F}_m^T & -\mathcal{T}_{m,A} Y_m(t) E_m(T_{m+1,m}^B)^T \\ -T_{m+1,m}^A E_m^T Y_m(t) \mathcal{T}_{m,B}^T & -T_{m+1,m}^A E_m^T Y_m(t) E_m(T_{m+1,m}^B)^T \end{bmatrix} \mathcal{W}_{m+1}^T.$$

Now, as $Y_m(t)$ exact solution of the low dimensional differential Stein equation

$$\dot{Y}_m(t) = \mathcal{T}_{m,A} Y_m(t) \mathcal{T}_{m,B}^T - Y_m(t) + \widetilde{E}_m \widetilde{F}_m^T,$$

so

$$R_m(t) = \mathcal{V}_{m+1} \begin{bmatrix} 0 & -\mathcal{T}_{m,A}Y_m(t)E_m(T^B_{m+1,m})^T \\ -T^A_{m+1,m}E^T_m Y_m(t)\mathcal{T}^T_{m,B} & -T^A_{m+1,m}E^T_m Y_m(t)E_m(T^B_{m+1,m})^T \end{bmatrix} \mathcal{W}^T_{m+1}.$$

and since $\mathcal{V}_{m+1}$ and $\mathcal{W}_{m+1}$ are orthonormal, we obtain

$$\|R_m(t)\|^2_F = \|T^A_{m+1,m}\overline{Y}_{m,1}(t)\mathcal{T}^T_{m,B}\|^2_F + \|T^B_{m+1,m}\overline{Y}_{m,2}(t)\mathcal{T}^T_{m,A}\|^2_F + \|T^A_{m+1,m}Y_{m,m}(t)(T^B_{m+1,m})^T\|_F.$$

where $\overline{Y}_{m,1}(t) = E^T_m Y_m(t)$, $\overline{Y}_{m,2}(t) = E^T_m Y_m(t)^T$ and $Y_{m,m}(t) = E^T_m Y_m(t)E_m$.  □

The approximate solution $X_m(t)$ can be given as a product of two matrices of low rank. Consider the singular value decomposition of the $2mr \times 2mr$ matrix:

$$Y_m(t) = \widetilde{Y}_1 \Sigma \widetilde{Y}^T_2,$$

where $\Sigma$ is the diagonal matrix of the singular values of $Y_m$ sorted in decreasing order. Let $Y_{1,l}$ and $Y_{2,l}$ be the $2mr \times l$ matrices of the first $l$ columns of $\widetilde{Y}_1$ and $\widetilde{Y}_2$, respectively, corresponding to the $l$ singular values of magnitude greater than some tolerance. We obtain the truncated singular value decomposition:

$$Y_m \approx U_{1,l} \Sigma_l U_{2,l}^T,$$

where $\Sigma_l = \text{diag}[\sigma_1,\dots,\sigma_l]$. Setting $Z_{1,m} = \mathcal{V}_m U_{1,l}\Sigma_l^{1/2}$, and $Z_{2,m} = \mathcal{W}_m U_{2,l}\Sigma_l^{1/2}$, it follows that:

$$X_m \approx Z_{1,m}(t) Z^T_{2,m}(t). \tag{7}$$

This is very important for large problems when one does not need to compute and store the approximation $X_m$ at each iteration.

The following result shows that the approximation $X_m(t)$ is an exact solution of a perturbed differential Stein equation.

**Theorem 2.** *Let $X_m(t)$ be the approximate solution given by* (2). *Then, we have*

$$\dot{X}_m(t) = (A - F_{m,A})X_m(t)(B - F_{m,B}) - X_m(t) + EF^T, \tag{8}$$

*where*

$$\begin{cases} F_{m,A} = V_{m+1}T^A_{m+1,m}E^T_m\mathcal{V}^T_m, \\ F_{m,B} = \mathcal{W}_m E_m(T^B_{m+1,m})^T W^T_{m+1}. \end{cases}$$

**Proof.** By multiplying the (5) left by $\mathcal{V}_m$ and right by $\mathcal{W}_m{}^T$, we obtain

$$\mathcal{V}_m \dot{Y}_m(t)\mathcal{W}^T_m = \mathcal{V}_m \mathcal{T}_{m,A}Y_m(t)\mathcal{T}^T_{m,B}\mathcal{W}^T_m - \mathcal{V}_m Y_m(t)\mathcal{W}^T_m + \mathcal{V}_m\widetilde{E}_m\widetilde{F}^T_m\mathcal{W}^T_m.$$

Using relationships

$$\begin{cases} A\mathcal{V}_m = \mathcal{V}_m\mathcal{T}_{m,A} + V_{m+1}T^A_{m+1,m}E^T_m, \\ \mathcal{W}^T_m B = \mathcal{T}^T_{m,B}\mathcal{W}^T_m + E_m(T^B_{m+1,m})^T W^T_{m+1}, \end{cases}$$

we find

$$\begin{aligned} \dot{X}_m(t) &= [A\mathcal{V}_m - V_{m+1}T^A_{m+1,m}E^T_m]Y_m(t)[\mathcal{W}^T_m B - E_m(T^B_{m+1,m})^T W^T_{m+1}] - X_m(t) + EF^T \\ &= [A - V_{m+1}(T^A_{m+1,m}E^T_m\mathcal{V}^T_m]X_m(t)[B - \mathcal{W}_m E_m(T^B_{m+1,m})^T W^T_{m+1}] - X_m(t) + EF^T. \end{aligned}$$

So

$$\dot{X}_m(t) = (A - F_{m,A})X_m(t)(B - F_{m,B}) - X_m(t) + EF^T,$$

where $F_{m,A} = V_{m+1} T^A_{m+1,m} V^T_m$ and $F_{m,B} = W_m (T^B_{m+1,m})^T W^T_{m+1}$. □

The next result states that the error $\mathcal{E}_m(t) = X(t) - X_m(t)$ satisfies also a differential Stein matrix equation.

**Theorem 3.** *Let $X(t)$, the exact solution of (1), and $X_m(t)$ be the low-rank approximate solution obtained at step m. The error $\mathcal{E}_m(t)$ satisfies the following differential Stein equation*

$$\dot{\mathcal{E}}_m(t) = A\mathcal{E}_m(t)B - \mathcal{E}_m(t) - R_m(t). \tag{9}$$

**Proof.** According to (1) and (4) we have

$$
\begin{aligned}
\dot{\mathcal{E}}_m(t) &= \dot{X}(t) - \dot{X}_m(t) \\
&= AX(t)B - X(t) + EF^T - AX_m(t)B + X_m(t) - EF^T - R_m(t) \\
&= A(X(t) - X_m(t))B - (X(t) - X_m(t)) - R_m(t) \\
&= A\mathcal{E}_m(t)B - \mathcal{E}_m(t) - R_m(t).
\end{aligned}
$$

□

Next, we give an upper bound for the norm of the error $X - X_m$ by using the 2-logarithmic norm defined by $\mu_2(A) = \frac{1}{2}\lambda_{\max}(A + A^T)$. The logarithmic norm satisfies the following relation

$$\| e^{tA} \|_2 \le e^{\mu_2(A)t}; \quad t \ge 0. \tag{10}$$

**Theorem 4.** *At step m of the extended block Arnoldi process, we have the following upper bound for the norm of the error,*

$$\|\mathcal{E}_m(t)\|_F \le e^{(t-t_0)\mu_2(\mathcal{A})} \|\mathcal{E}_m(t_0)\|_F + \max_{\tau \in [t_0, t]} \|R_m(\tau)\|_F \frac{e^{(t-t_0)\mu_2(\mathcal{A})} - 1}{\mu_2(\mathcal{A})}.$$

**Proof.** We notice that the differential Stein Equation (9) is equivalent to the linear ordinary differential equation

$$
\begin{cases}
\dot{err}_m(t) = \mathcal{A}err_m(t) - b_m(t), \\
err_0 = vec(\mathcal{E}_m(t_0)),
\end{cases} \tag{11}
$$

where

$$
\begin{cases}
\mathcal{A} = B^T \otimes A - I_{np}, \\
err_m(t) = vec(\mathcal{E}_m(t)), \\
b_m(t) = vec(R_m(t)),
\end{cases}
$$

Then, the error $err_m(t)$ can be expressed in the integral form as follows

$$err_m(t) = e^{(t-t_0)\mathcal{A}} err_0 - \int_{t_0}^t e^{(t-\tau)\mathcal{A}} b_m(\tau) d\tau, \quad t \in [t_0, T_f].$$

By using $\|e^{t\mathcal{A}}\|_2 \leq e^{\mu_2(\mathcal{A})t}$, we have

$$
\begin{aligned}
\|err_m(t)\|_2 &\leq \|e^{(t-t_0)\mathcal{A}}err_0 - \int_{t_0}^t e^{(t-\tau)\mathcal{A}}b_m(\tau)d\tau\|_2 \\
&\leq \|e^{(t-t_0)\mathcal{A}}err_0\|_2 + \|\int_{t_0}^t e^{(t-\tau)\mathcal{A}}b_m(\tau)d\tau\|_2 \\
&\leq e^{(t-t_0)\mu_2(\mathcal{A})}\|err_0\|_2 + \int_{t_0}^t e^{(t-\tau)\mu_2(\mathcal{A})}\|b_m(\tau)\|_2 d\tau \\
&\leq e^{(t-t_0)\mu_2(\mathcal{A})}\|err_0\|_2 + \max_{\tau\in[t_0,t]}\|b_m(\tau)\|_2 \int_{t_0}^t e^{(t-\tau)\mu_2(\mathcal{A})}d\tau \\
&\leq e^{(t-t_0)\mu_2(\mathcal{A})}\|err_0\|_2 + \max_{\tau\in[t_0,t]}\|b_m(\tau)\|_2 \frac{e^{(t-t_0)\mu_2(\mathcal{A})}-1}{\mu_2(\mathcal{A})} \\
&\leq e^{(t-t_0)\mu_2(\mathcal{A})}\|err_0\|_2 + \max_{\tau\in[t_0,t]}\|vec(R_m(\tau))\|_2 \frac{e^{(t-t_0)\mu_2(\mathcal{A})}-1}{\mu_2(\mathcal{A})}.
\end{aligned}
$$

As $\|vec(\mathcal{E}_m(t))\|_2 = \|\mathcal{E}_m(t)\|_F$, so

$$
\|\mathcal{E}_m(t)\|_F \leq e^{(t-t_0)\mu_2(\mathcal{A})}\|\mathcal{E}_m(t_0)\|_F + \max_{\tau\in[t_0,t]}\|R_m(\tau)\|_F \frac{e^{(t-t_0)\mu_2(\mathcal{A})}-1}{\mu_2(\mathcal{A})}.
$$

$\square$

## 3. Solving the Projected Differential Stein Matrix Equation

### 3.1. Rosenbrock Method

In this section, we are applying the Ros method [18] for solving the projected differential Stein matrix equation. We can write the low dimensional nonsymmetric differential Stein Equation (5) in the following form

$$
\begin{cases}
\dot{Y}_m(t) = \mathcal{S}(Y_m(t)), & t \in [t_0, T_f] \\
Y_m(t_0) = Y_m^{(0)},
\end{cases}
\tag{12}
$$

where $\mathcal{S}(Y_m) = \mathcal{T}_{m,A}Y_m\mathcal{T}_{m,B}^T - Y_m + \widetilde{E}_m\widetilde{F}_m^T$ and $Y_m^{(0)} = \mathcal{V}_mX_0\mathcal{W}_m^T$.

The approximation $Y_{m,k+1}$ of $Y_m(t_{k+1})$ obtained at step $k+1$ by Ros method is given by

$$
Y_{k+1} = Y_k + \frac{3}{2}P_1 + \frac{1}{2}P_2,
\tag{13}
$$

where $P_1$ and $P_2$ solve the following Stein matrix equations

$$
\Lambda_A P_1 \Lambda_B^T - P_1 + \mathcal{S}(Y_k) = 0,
\tag{14}
$$

and

$$
\Lambda_A P_2 \Lambda_B^T - P_2 + \mathcal{S}(Y_k + P_1) - \frac{2}{h}P_1 = 0,
\tag{15}
$$

where

$$
\begin{cases}
\Lambda_A = \sqrt{\gamma}\mathcal{T}_{m,A} + \sqrt{\frac{1}{h}+1-\gamma}I_{size(\mathcal{T}_{m,A})}, \\
\Lambda_B = \sqrt{\gamma}\mathcal{T}_{m,B} + \sqrt{\frac{1}{h}+1-\gamma}I_{size(\mathcal{T}_{m,B})},
\end{cases}
$$

with $\gamma = 1/2$.

The Ros algorithm for solving the reduced differential Stein matrix Equation (5) is summarized in Algorithm 2.

---

**Algorithm 2:** The 2-Rosenbrock method for solving the reduced NDSE (12)

---

**Input**: $\mathcal{T}_{m,A}, \mathcal{T}_{m,B}, \widetilde{E}_m, \widetilde{F}_m, t_0, T_f$.

1.  Choose $h$.
2.  Compute: $N = \frac{T_f - t_0}{h}$
3.  Compute: $\Lambda_A$
4.  Compute: $\Lambda_B$
5.  For $k = 1 : N + 1$

    (a)    Compute $P_1$ by solving Stein matrix Equation (14).
    (b)    Compute $P_2$ by solving Stein matrix Equation (15).
    (c)    Compute $Y_{k+1}$ by (13).

6.  End For $k$.

**Output**: $Y_{T_f}$.

---

### 3.2. Backward Differentiation Formula Method

In this section, we present a BDF method [19] for solving, at each step $m$ of the extended block Arnoldi process, the low dimensional differential Stein matrix Equation (5).

At each time $t_k$, let $Y_{m,k}$ be the approximation of $Y_m(t_k)$, where $Y_m$ is a solution of (5). Then, the new approximation $Y_{m,k+1}$ of $Y_m(t_{k+1})$ obtained at step $k+1$ by BDF method is defined by the implicit relation

$$Y_{m,k+1} = \sum_{i=0}^{p-1} \alpha_i Y_{m,k-i} + h\beta \mathcal{S}(Y_{m,k+1}), \tag{16}$$

where $h = t_{k+1} - t_k$ is the step size, $\alpha_i$ and $\beta$ are the coefficients of the BDF method as listed in the Table 1.

**Table 1.** Coefficients of the $p$-step BDF method with $p \leq 2$.

|        | $p$ | $\beta$ | $\alpha_0$ | $\alpha_1$ |
|--------|-----|---------|------------|------------|
| BDF1   | 1   | 1       | 1          | 0          |
| BDF2   | 2   | 2/3     | 4/3        | −1/3       |

The approximate $Y_{m,k+1}$ solves the following matrix equation

$$-Y_{m,k+1} + h\beta(\mathcal{T}_{m,A} Y_{m,k+1} \mathcal{T}_{m,B}^T - Y_{m,k+1} + \widetilde{E}_m \widetilde{F}_m^T) + \sum_{i=0}^{p-1} \alpha_i Y_{m,k-i} = 0.$$

We assume that at each time $t_k$, the approximation $Y_{m,k}$ is factorized as a low-rank product $Y_{m,k} \approx \widetilde{U}_{m,k} \widetilde{V}_{m,k}^T$, where $\widetilde{U}_{m,k} \in \mathbb{R}^{n \times m_k}$ and $\widetilde{V}_{m,k} \in \mathbb{R}^{p \times m_k}$, with $m_k \ll n, p$. We define

$$\begin{cases} \widetilde{\mathbb{T}}_{m,A} = \sqrt{h\beta}\mathcal{T}_{m,A} - \sqrt{h\beta}I_{2m}, \\ \widetilde{\mathbb{T}}_{m,B} = \sqrt{h\beta}\mathcal{T}_{m,B}^T + \sqrt{h\beta}I_{2m}, \\ \mathbb{E}_{m,k+1} = [\sqrt{h\beta}\widetilde{E}_m^T, \sqrt{\alpha_0}\widetilde{U}_{m,k}^T, ..., \sqrt{\alpha_{p-1}}\widetilde{U}_{m,k+1-p}^T]^T, \\ \mathbb{F}_{m,k+1} = [\sqrt{h\beta}\widetilde{F}_m^T, \sqrt{\alpha_0}\widetilde{V}_{m,k}^T, ..., \sqrt{\alpha_{p-1}}\widetilde{V}_{m,k+1-p}^T]^T. \end{cases}$$

Then, we obtain the following matrix Stein equation:

$$\widetilde{\mathbb{T}}_{m,A} Y_{m,k+1} \widetilde{\mathbb{T}}_{m,B} - Y_{m,k+1} + \mathbb{E}_{m,k} \mathbb{F}_{m,k}^T = 0. \tag{17}$$

The low-rank approximate solutions method by extended block Arnoldi algorithm for the large differential nonsymmetric stein matrix equations is summarized as follows in Algorithm 3.

---

**Algorithm 3:** The low-rank extended block Arnoldi method for DNSE

---

**Input**: $X_0$, choose a tolerance *tol* $> 0$, a maximum number of $m_{max}$ iterations.
1.     For $m =, 2, 3, ..., m_{max}$ do
2.     Update $\mathcal{V}_m, \mathcal{T}_{m,A}$, by Algorithm 1 (EBA) applied to $(A, E)$
3.     Update $\mathcal{W}_m, \mathcal{T}_{m,B}$, by Algorithm 1 (EBA) applied to $(B^T, F)$
4.     Solve the low-dimensional problem (5) by BDF method or Ros method.
5.     If $\|R_m(t)\|_F < tol$, stop.
6.     End If;
7.     End For $(m)$;
8.     Using (7), the approximate solution $X_m$ is given by $X_m \approx Z_{1,m} Z_{2,m}^T$.

**Output**: $X_m$.

---

## 4. Numerical Experiments

In this section, we give some numerical examples of large nonsymmetric differential Stein matrix equations. All the experiments were performed on a computer of Intel Core i5 at 1.3 GHz and 4 GB of RAM. The Algorithm 3 were coded in Matlab2014. In all of the examples, the coefficients of the matrices $E$ and $F$ were random values uniformly distributed on $[0, 1]$. The stopping criterion used for EBA-BDF method and EBA-Ros was $\|\mathcal{R}(X_m)\|_F < 10^{-10}$ or a maximum of $m_{max} = 40$ iterations was achieved.

### 4.1. Example 1

In this first example, the matrices $A$ and $B$ are obtained from the centered finite difference discretization of the operators:

$$L_A(u) = \Delta u + f_1(x, y) \frac{\partial u}{\partial x} +, f_2(x, y) \frac{\partial u}{\partial y} + f(x, y)u,$$

$$L_B(u) = \Delta u + g_1(x, y) \frac{\partial u}{\partial x} + g_2(x, y) \frac{\partial u}{\partial y} + g(x, y)u,$$

on the unit square $[0, 1] \times [0, 1]$ with homogeneous Dirichlet boundary conditions. The number of inner grid points in each direction was $n_0$ and $p_0$ for the operators $L_A$ and $L_B$, respectively. The matrices $A$ and $B$ were obtained from the discretization of the operator $L_A$ and $L_B$ with the dimensions $n = n_0^2$ and $p = p_0^2$, respectively. The discretization of the operator $L_A(u)$ and $L_B(u)$ yields matrices extracted from the `Lyapack` package [20] using the command `fdm_2d_matrix` and denoted as A = fdm($n_0$,'f_1(x,y)','f_2(x,y)','f(x,y)'). In this example, $n = 10,000$ and $p = 10,000$, respectively, and are named as $A = $ fdm($n_0, f_1(x, y), f_2(x, y), f(x, y)$) and $B = $ fdm($p0, g_1(x, y), g_2(x, y), g(x, y)$) with $f_1(x, y) = -e^{xy}, f_2(x, y) = -\sin(xy), f(x, y) = y^2, g_1(x, y) = -100e^x, g_2(x, y) = -12xy$ and $g(x, y) = \sqrt{x^2 + y^2}$. For this experiment, we used $r = 2$.

In Figure 1, we plotted the Frobenius norms of the residuals versus the number of iterations for the EBA-BDF and the EBA-Ros method.

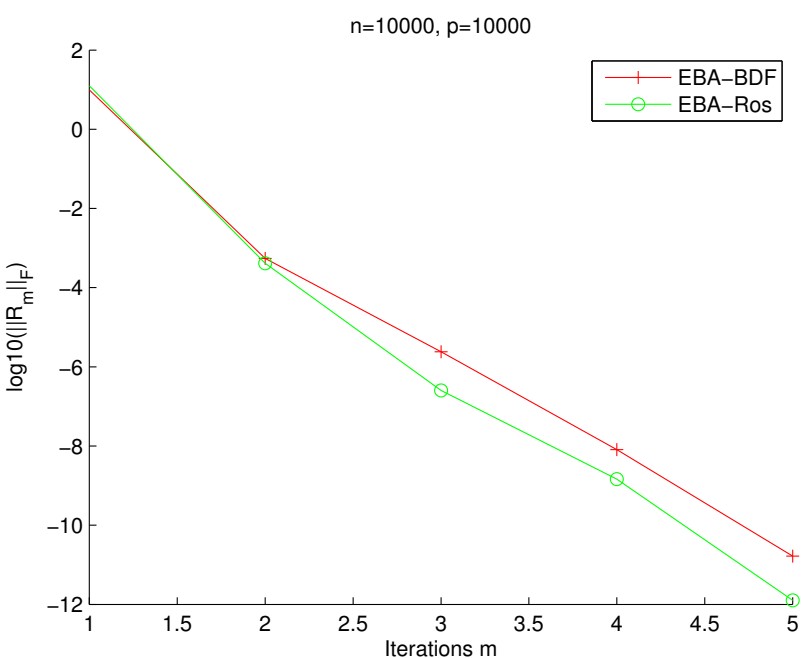

**Figure 1.** EBA-BD F method and EBA-Ros method.

In Table 2, we compared the performances of the EBA-BDF method and the EBA-Ros. For both methods, we listed the residual norms, the maximum number of iteration and the corresponding execution time.

**Table 2.** Results for Example 1.

| Test Problem | Method | Iterations | Residual Norm | Times (s) |
|---|---|---|---|---|
| $n = 8100, p = 4900, r = 2, h = 0.3$ | EBA-BDF | 5 | $2.03 \times 10^{-11}$ | 0.497 |
| | EBA-Ros | 5 | $1.40 \times 10^{-11}$ | 0.854 |
| $n = 10{,}000, p = 4900, r = 3, h = 0.2$ | EBA-BDF | 5 | $4.38 \times 10^{-12}$ | 1.025 |
| | EBA-Ros | 5 | $5.77 \times 10^{-13}$ | 1.53 |
| $n = 40{,}000, p = 12{,}100, r = 4, h = 0.1$ | EBA-BDF | 5 | $8.27 \times 10^{-12}$ | 6.80 |
| | EBA-Ros | 5 | $8.70 \times 10^{-13}$ | 6.85 |

*4.2. Example 2*

For the second set of experiments, we use the matrices *add*32, *pde*2961, and *thermal* from the Harwell Boeing Collection [21]. The tolerance was set to $10^{-8}$ for the stop test on the residual. For the EBA-BDF and EBA-Ros methods, we used a constant timestep $h = 0.1$,

In Figure 2, the matrices $A = add32$ and $B = pde2961$ with dimensions $n = 4960$ and $p = 2961$, respectively, and $r = 2$. We plotted the Frobenius norms of the residuals $\|R_m(T_f)\|_F$ at final time $T_f$ versus the number of extended block Arnoldi iterations for the EBA-BDF and EBA-Ros method.

In Figure 3, the matrices $A = \mathtt{fdm}(x + 10y^2, \sqrt{2x^2 + y^2}, y^2 - x^2)$ and $B = thermal$ with dimensions $n = 10{,}000$ and $p = 3456$, respectively, $h = 0.2$ and $r = 3$. We plotted the Frobenius norms of the residuals $\|R_m(T_f)\|_F$ at final time $T_f$ versus the number of extended block Arnoldi iterations for the EBA-BDF and EBA-Ros method.

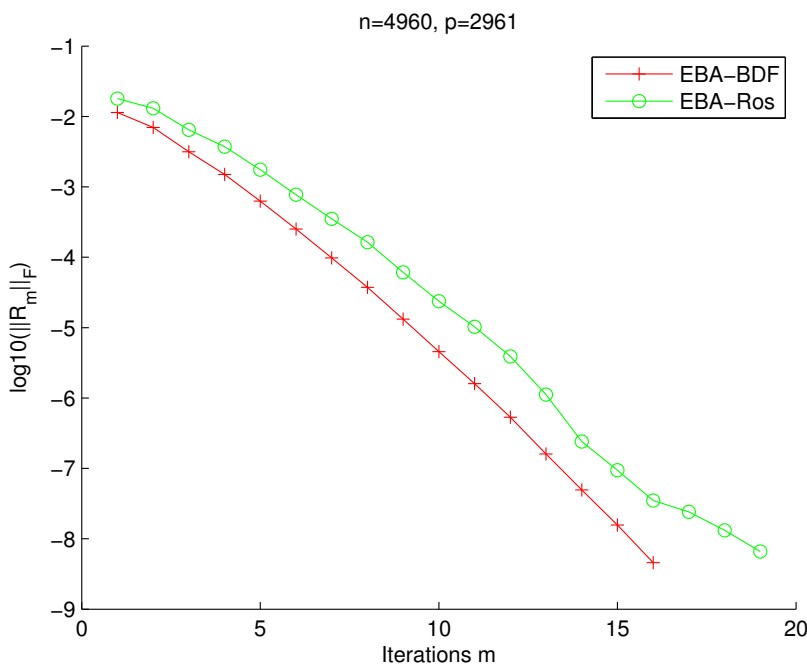

**Figure 2.** Residual norm vs number *m* of extended block Arnoldi iterations.

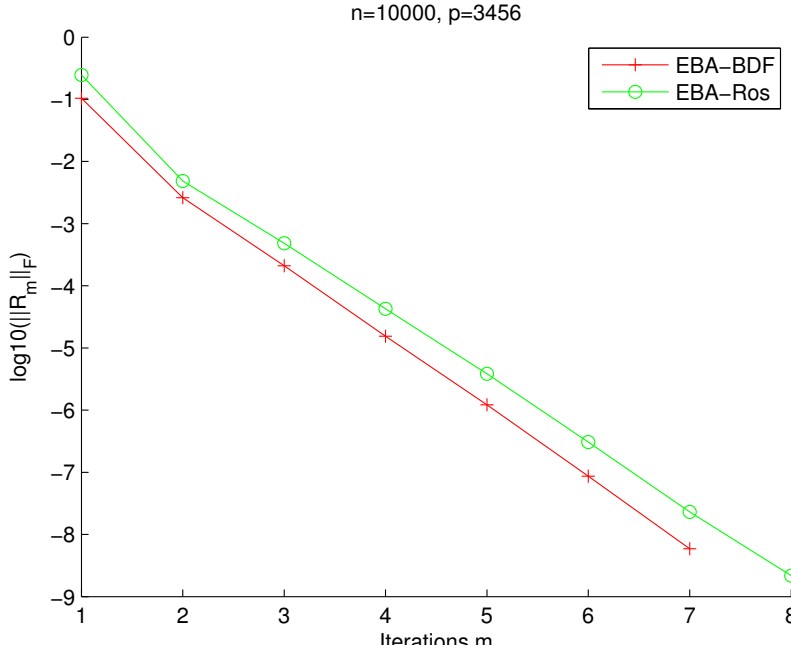

**Figure 3.** Residual norm vs number *m* of extended block Arnoldi iterations.

In Table 3, we list the Frobenius residual norms at final time $T_f = 2$ and the corresponding CPU time for each method.

**Table 3.** Runtimes in seconds and the residual norms.

| Test Problem | Method | CPU Time | Iterations | $\|R_m(T_f)\|_F$ |
|---|---|---|---|---|
| $n = p = 2500$ $r = 2$ and $h = 0.2$ | EBA-BDF | 11.01 s | 16 | $4.57 \times 10^{-9}$ |
| $A = add32$ and $B = pde2961$ | EBA-Ros | 11.49 s | 19 | $6.59 \times 10^{-9}$ |
| $n = 12,100$, $p = 4900$ | EBA-BDF | 10.77 s | 4 | $7.75 \times 10^{-9}$ |
| $p = 8100$, $h = 0.1$ and $r = 3$ | EBA-Ros | 10.78 s | 5 | $6.31 \times 10^{-10}$ |
| $B = thermal$, $n = 10,000$, $p = 3456$ and $r = 3$, $h = 0.2$ | EBA-BDF | 6.20 s | 7 | $5.87 \times 10^{-9}$ |
| $A = \mathtt{fdm}(x + 10y^2, \sqrt{2x^2 + y^2}, y^2 - x^2)$ | EBA-Ros | 6.27 s | 8 | $2.18 \times 10^{-9}$ |

The numerical results are promising, showing the effectiveness of the proposed methods.

## 5. Conclusions

We presented, in this paper, a new iterative method for computing numerical solutions for large-scale differential Stein matrix equations with low-rank right-hand sides. This approach is based on projection onto extended block Krylov subspaces with a Galerkin method. The approximate solutions are given as products of two low-rank matrices and allow for saving memory for large problems. The numerical experiments show that the proposed extended block Krylov-based method is effective for large and sparse matrices.

**Author Contributions:** Conceptualization, L.S., E.M.S. and H.T.A.; methodology, E.M.S.; software, L.S.; validation, L.S., E.M.S. and H.T.A.; formal analysis, L.S.; investigation, E.M.S.; writing—original draft preparation, L.S., E.M.S. and H.T.A.; writing—review and editing, L.S., E.M.S. and H.T.A. All authors have read and agreed to the published version of the manuscript.

**Funding:** This research received no external funding.

**Acknowledgments:** The authors should express their deep-felt thanks to the anonymous referees for their encouraging and constructive comments, which improved this paper.

**Conflicts of Interest:** The authors declare no conflict of interest.

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
