# Peer review of "On Some Numerical Methods for Solving Large Differential Nonsymmetric Stein Matrix Equations"

_mca, doi:10.3390/mca27040069_

Round 1

Reviewer 1 Report

Report for article  

" On some numerical methods for Solving large differential nonsymmetric Stein matrix equations"

This manuscript deals with a new numerical method based on the extended block Arnoldi algorithm for solving large scale differential nonsymmetric Stein matrix equations with low rank right hand sides.  

I am happy to recommend the present paper for publication. Nevertheless, before publishing, I also have quite a few technical inquiries:

1-  There is an extra (missing) endpoint, comam. There might be more typos like this. So, the authors should proofread the final version carefully.

2-  We note the presence of syntax errors in English language, please made a careful revision.

3-  It will be better to improve the introduction and add some  recent relater paper like:  DOI: 10.5269/bspm.46032 and DOI: 10.5269/bspm.46315  and DOI: 10.2298/TAM201123013G and DOI: 10.1007/s11587-021-00649-2

The paper is correct and the results deserve to be published  in MCA.

Author Response

Dear Professor,

In the revised version of the paper ”On some numerical methods for Solving large differential nonsymmetric Stein matrix equations”, submitted for publication in the journal Mathematical and Computational Applications. We would like to thank again the reviewers for taking their time to read carefully the manuscript and for their valuable comments and useful corrections. We have modified the paper according to the Referees remarks and we list here the principal modifications. The main changes in the paper are in blue  and some of them are listed in the attached file.

Please find herewith attached my revised version to Mathematical and Computational Applications Journal.

Thank you very much.

Sincerely yours,

EL. M. SADEK

Reviewer 2 Report

See the attachment.

Author Response

(The authors gave the same response as above.)

Reviewer 3 Report

See the attached PDF file.

Author Response

(The authors gave the same response as above.)

Reviewer 4 Report

The aim of this manuscript is the study of " On some numerical methods for Solving large differential nonsymmetric Stein matrix equations"and the goal of this manuscript is to propose a new numerical method based on the extended block Arnoldi algorithm for solving large scale differential nonsymmetric Stein matrix equations with low rank right hand sides. These algorithm is based a new method to solve differential nonsymmetric.

The paper is organized as follows.

Section 1 is Introduction with a description of the importance of the notions.

Section 2, The Extended Block Arnoldi Algorithm, is dedicated to  the extended block Arnoldi algorithm(EBA). 

In Section 3 Low-rank Approximate Solutions Method are provided  four new  theorems. .

Section 4 contains the Rosenbrock Method and Backward Differentiation Formula Method.

In section 5 are given some Numerical Experiments of large nonsymmetric differential  Stein matrix equations. 

Conclusions are presented in Section 6 and Section 7 is dedicated to References.

I recommend to the authors to have a Section 2 with Preliminaries. This section is assumed to be dedicated to more definitions and results, which are needed to understand the new results.

Also, I recommend to the authors to use Ym(t) instead of Ym, in Section 3, Theorem 1. For example in the relation of the last row from the third page, right after proof.

The paper is well organized and written and it certainly brings an outcome to the field. The mathematical means are correct. The wording is carefully done.

I recommend the publication of the manuscript.

Author Response

(The authors gave the same response as above.)

Round 2

Reviewer 3 Report

See the attached PDF file.

Author Response

Dear Professor,

First of all, we would like to thank you very much for taking their time to read the manuscript carefully and for you valuable comments. We have modified the manuscript following referee suggestions. We list here the principal modifications given in the revised version of the paper ”On some numerical methods for Solving large differential nonsymmetric Stein matrix equations”. The main changes are in blue.

Please find herewith attached my revised version submission to Mathematical and Computational Applications Journal.

Thank you very much.
Sincerely yours,
EL. M. SADEK
